# A Non-Invasive Investigation into the Prevalence of Higher than Normal Blood Pressure, Hypertension and the Association between Blood Pressure and Body Weight in Male and Female Adolescents in the Polokwane Local Municipality, Limpopo-South Africa: A Cross-Sectional Study

**DOI:** 10.3390/children7030018

**Published:** 2020-03-04

**Authors:** Thato Tshepo Raphadu, Marlise Van Staden, Winnie Maletladi Dibakwane, Kotsedi Daniel Monyeki

**Affiliations:** Department of Physiology and Environmental Health, University of Limpopo, Sovenga 0727, South Africa; thatoraphadu@gmail.com (T.T.R.); marlise.vanstaden@ul.ac.za (M.V.S.); winnie.dibakwane@ul.ac.za (W.M.D.)

**Keywords:** adolescent, pre-hypertension, hypertension, overweight, obesity, Polokwane

## Abstract

Background: Hypertension (HT) is one of the greatest non-communicable diseases affecting both sexes in all human populations; and it is a major cause of death and morbidity across the world. The purpose of this study was to investigate the prevalence of pre-hypertension, hypertension and investigate the association between blood pressure (BP) and body weight (looking into body mass index (BMI) and body surface area (BSA)). Method: A cross-sectional study of adolescents aged 13–19 years was conducted at three secondary schools consisting of 121 females and 97 males. Data on height; weight; and blood pressure was collected from all participants. BMI and BP percentiles were calculated for each participant. Results: The prevalence of pre-hypertension and hypertension in males was 28.9% and 10.3% compared to 25.6% and 7.4% in females. The prevalence of pre-hypertension and hypertension in adolescents who are overweight/obese was 7.3% and 2.7%. A weak positive association between BMI; systolic blood pressure (SBP) and diastolic blood pressure (DBP) was found (*r* = 0.254 and 0.216; *p*-value = 0.001) for the whole population. A significant, moderate correlation was found between BSA on SBP was found (*r* = 0.407, *p*-value = 0.001); and a significant, weak correlation between BSA and DBP was found (*r* = 0.183, *p*-value = 0.007. In conclusion, the prevalence of pre-hypertension and hypertension was higher in males as compared to females. Results highlight the need for screening for blood pressure and body weight in adolescents; especially in adolescents who were found to have pre-hypertension as they are at high risk of developing hypertension.

## 1. Introduction

The WHO defines non-communicable diseases (NCDs) as conditions that are influenced by environmental, behavioral, physiological and genetical factors [1]. The burden of NCD is drastically increasing worldwide. It is estimated that NCDs kills 41 million people every year, which is equivalent to 71% of deaths worldwide [1]. Deaths caused by NCDs are expected to increase by 15% between 2010 and 2020 [2]. There are a variety of NCDs such as type 2 diabetes mellitus, obesity, cardiovascular diseases (CVDs), hypertension and diseases associated with smoking, drug abuse and alcohol.

Hypertension in adolescents is defined as systolic and/or diastolic blood pressure greater than the 95th percentile for height, sex and age. Normal BP is defined as below the 90th percentile, whereas BP above the normal BP is defined to be between 90th and 95th percentile for height, sex and age but exceeds 120/80 mmHg [3]. The diagnosis of high BP in adolescents is rare due to the lack of inclusion of BP measurements during routine screening for hypertension, as the focus of hypertension screening has always been on adults and not on children and adolescents [4]. Hypertension is complex and a multifactorial disease that makes it difficult to diagnose. Risk factors that are mainly associated with hypertension are overweight and obesity in both children and adolescents [4].

High blood pressure is the most common disease identified in overweight adolescents, and persistent high blood pressure can lead to mortality in adulthood. High blood pressure has been attributed to approximately 12.8% of the deaths worldwide [5]. Several studies have investigated the link between BMI and BP in the pediatric and adolescent population, with most investigations finding the correlation coefficient to be greater than 0.3 which showed an association of a moderate to strong link [5]. Weight gain is usually linked with a corresponding increase in BP in children and adolescents [4]. Early diagnosis of high BP will result in early treatment, reducing or preventing further complications associated with high BP. However, information is available on the prevalence of BP in so many countries including South Africa. At present, there is a shortage of literature relating the link between BP and body weight in adolescent learners in the Polokwane region, South Africa. It is however, of a great significant to obtain more information regarding the prevalence and the possible association between BP and body weight (Particularly BMI and BSA), since it plays an important role in various aspects of children’s development, as already indicated. Thus, the study aims to determine the prevalence of pre-hypertension, hypertension and investigate the possible association between BP and body weight (looking into BMI and BSA).

## 2. Materials and Methods

### 2.1. Sampling Procedure

This is a cross-sectional study, in which data was collected among urban and rural secondary-school participants, based in the Polokwane region of the Limpopo province, which is located the north of South Africa. The province is still under-resourced compared with other provinces in South Africa. It is characterized by high levels of unemployment, reflecting a transition between traditional and western lifestyles since the majority of the young adults work in towns and cities and return home for periodic visits [6,7]. Three (3) secondary schools were selected from the region by the department of basic education along with the school principals. For the purpose of this analysis, a portion of the sample was recruited. In total, 218 adolescents (121 females and 97 males), aged between 13 to 19 years. The University of Limpopo Ethics Committee and the Limpopo province department of education approved the study in 2015. The reference number for this study is “Project Identification ID: REC-0310111-031. Permission to carry out the study was granted by the provincial heads of the department of education, the district managers for the department of basic education in Limpopo province, and the principals of schools. All the parents/guardians of the subjects were informed about the objectives and procedures and gave informed consent for their subjects to participate. Participants gave assent before allowing them to participate in the study. Confidentiality and anonymity of all the results were assured.

### 2.2. Exclusion and Inclusion Criteria

Adolescents with any of the following conditions were excluded from the study:Adolescents with cardiovascular conditions.Females who are pregnant or breastfeeding.Adolescents taking acute or chronic medication.Adolescents with orthopedic injuries.Adolescents who are using supplements (e.g., steroids).

### 2.3. Blood Pressure

Blood pressure was measured with an Omron blood pressure device as described by the American Heart Association [8]. The cut–off points for high blood pressure or hypertension (Table 1) as described by NHLBI (2005), were utilized. For adolescents aged eighteen to nineteen years, pre-hypertension was defined as average SBP/DBP greater or equal to 120/80 mmHg, whereas hypertension was defined as average SBP/DBP greater or equal to 140/90 mmHg [9].

### 2.4. Anthropometric Data (Weight and Height)

The field workers were trained to collect anthropometric data. All study participants underwent a series of anthropometric measurements (height, and weight and waist circumference) according to standard procedures of the International Society for the Advancement of Kinanthropometry [11]. The weights and heights were measured using a calibrated electronic scale to the nearest 0.1 kg, and Leicester portable stadiometer equipment to the nearest 0.1 cm. A flexible steel tape was used to measure the waist circumference (WC) to the nearest 0.1 cm, midway between the last rib and the iliac crest, with the participants standing. Body mass index (BMI) was calculated as weight (kg) per height squared (m^2^). Body mass index cut-offs was as follows: below 18.5 kg/m^2^—underweight, 18.5 kg/m^2^ to 24.99 kg/m^2^—normal, 25 kg/m^2^ to 29.99 kg/m^2^—overweight and above 30 kg/m^2^—obese [12]. Body surface area was measured using the Du Bois formula as follows:BSA = 0.20247 × Height (m)^0.725^ × Weight (Kg)^0.425^

Due to the lack of previous data assessing the impact of BSA on BP, we categorized BSA using quartiles to report more useful findings on the association of BSA with BP. Participants were classified as obese or non-obese by using BSA median values of BSA ≥ 1.6 (obese) or BSA < 1.6 (non-obese) [13]. Each participant was requested to present themselves barefoot, in shorts and T-shirts for the anthropometric measurements.

### 2.5. Statistical Analysis

IBM SPSS Version 25 was utilized to conduct statistical analysis. Distribution frequencies were calculated for all variables, and any outlier was verified against the raw data and where necessary, the outlier was removed. To describe and characterize the samples, descriptive statistics were calculated for all the variables in order to indicate frequencies (expressed as percentages), means and standard deviations. This served as an indication of prevalence. The estimation of prevalence was presented by sex. The association between BP and anthropometric data (BMI and BSA) was tested using the Pearson correlation. Participants were divided into different age and sex categories for further analysis of the results. The probability value for statistical significance for all tests was set at a *p*-value of <0.05.

### 2.6. Reliability and Validity

Measurements that were taken for both anthropometric (weight and height) and blood pressure were repeated throughout the study. If the first measurement differed from the second measurement by more than 5% for anthropometric measurements or 5 mmHg for blood pressure, a third measurement was taken to ensure the accuracy of the measurements. An average of the two closest measurements taken was utilized.

## 3. Results

### 3.1. Characteristics of the Population

A complete BP and anthropometric data were obtained from 218 participants (121 females and 97 males). The average age and mean systolic blood pressure for males was significantly higher than in females, whereas mean diastolic was significantly higher in females compared to males. Females had a heavier weight than males, whereas males were taller than females. Females also had a significantly higher BMI than in males, whereas the mean average for BSA was the same for both sexes (Table 2).

### 3.2. Prevalence of Prehypertension, Hypertension, Overweight and Obesity

The prevalence of pre-hypertension and hypertension was 28.9% and 10.3% in males whereas in females it was 25.6% and 7.4%. The overall prevalence of pre-hypertension and hypertension was 27.3% and 17.1%, respectively (Figure 1). The prevalence of overweight and obesity was 19.8% and 9.9% in females whereas in males it was 3.1% and 1.0%. The overall prevalence of overweight and obesity was 11.5% and 5.5%, respectively (Figure 2).

### 3.3. Prevalence of Pre-Hypertension and Hypertension in Adolescents Who Were Overweight/Obesity

The prevalence of pre-hypertension in adolescents who were overweight and obese was 3.2% and 4.1%. The prevalence of hypertension in adolescents who were overweight and obese was 2.3% and 0.4%. The combined prevalence for pre-hypertension of overweight and obesity was 7.3%, whereas the combined prevalence for hypertension of overweight and obesity was 2.7%, respectively (Table 3).

### 3.4. The Effect of BMI on SBP and DBP

A significant weak positive correlation of BMI on SBP (Figure 3) and BMI on DBP (Figure 4) was found (r = 0.254, r = 0.216, both with a *p*-value = 0.001).

### 3.5. The Effect of BSA on SBP and DBP

A significant, moderate positive correlation was found between BSA on SBP (Figure 5) was found (*r* = 0.407, *p*-value = 0.001); and a significant, weak positive correlation between BSA and DBP (Figure 6) was found (*r* = 0.183, *p*-value = 0.007).

## 4. Discussion

This was a cross-sectional study that investigated the prevalence of pre-hypertension and hypertension and aimed to explore the possible association between blood pressure and body weight among male and female adolescents in the Polokwane local municipality. The results obtained indicated a significant difference in BP between males and females’ adolescents. A positive association was found between BMI and BP (SBP and DBP); and between BSA and BP (SBP and DBP). The results showed that females had a significantly higher weight/BMI than males subsequently; the prevalence of overweight and obesity was higher in females compared to males. The relationship between overweight/obesity and hypertension has long been investigated, with the risk of hypertension being up to five times higher among obese people than among those of normal weight [14].

A study by Pate et al. [15] indicated that females progressed in puberty earlier than their male counterparts, in which they tend to be less physical active and prefer to be indoors rather than outdoors. This trend describes why females tend to be physically bigger compared to males of their same age. The schools that were used for this study were situated in a peri-urban area, which is rapidly undergoing transformation in lifestyle and eating habits. Many townships have an increase in the availability of fast food, this leads to adolescents being exposed to a high energy dense food that increases the risk of adolescents being obese. The prevalence of overweight and obesity was lower compared to the study conducted in Mthatha by Nkeh-Chungag et al. [16], which showed a prevalence of 33.6% for overweight and 31.1% for obesity. This might be due to the difference in geographical areas, culture or beliefs of adolescents and a different exposure to different types of food.

The prevalence of pre-hypertension and hypertension was higher in males compared to females, this shows a similar finding compared to the study conducted in Mthatha, South Africa [16]. The reason for a higher prevalence in males than females has not be made clear, but some studies have speculated that testosterone in males may play a role in the increase of blood pressure in male adolescents as compared to females [17], whereas in females estrogen and estrogen receptors stimulation have protective effects on the cardiovascular system that decrease the incidence of CVDs [18,19]. Respectively, the prevalence of pre-hypertension was higher, and the prevalence of hypertension was lower compared to the prevalence obtained by Nkeh-Chungag et al. [16]. This may also be due to geographical differences and the differences in the sample size, as this study was conducted at Mthatha, Eastern Cape.

A positive association of BMI with SBP and DBP for the whole population was found; these results confirm the findings obtained by Erlingsdottir et al. [20] which indicated an association between overweight/obesity and an increase in blood pressure. The Nkeh-Chungag et al. [16] study found a mean significant and weak positive correlation between BMI and BP (SBP and DBP) ((*r* = 0.198, *p*-value = 0.001); (*r* = 0.128, *p*-value = 0.010)). A study conducted in the western Africa, Nigeria indicated a significant and a weak positive correlation between BMI and BP (SBP and DBP) ((*r* = 0.142, *p*-value < 0.001); (*r* = 0.099, *p*-value = 0.010)) [21]; and similar study conducted in the middle east, United Emirates Arabs showed similar results between the correlation of BMI and SBP (*r* = 0.255, *p*-value < 0.01); and between BMI and DBP (*r* = 0.175, *p*-value < 0.01) [22]. Our results agree and are consistent with those of numerous other studies showing that BMI and BP levels are positively correlated. Although there is lack of research that has been done on the association of BSA on BP, we have found a significant positive (moderate and strong) association between BSA and BP (SBP and DBP). This indicates a need to further investigate the association of BSA on BP. This will help to compare and expand on the effect of BMI and BSA on BP.

However, it is still unclear of what could cause an elevation in blood pressure; some studies suggest that the mechanism that causes an increase in blood pressure may be due to pro-inflammatory immune responses in the cytokines and an increase in adipocyte secretion of adipokines which physiological dysfunction leading to an increase in blood Pressure [20]. A number of pathways have been described, with recent research focusing more on the neurohormonal aspect. The interrelation between those mechanisms is in itself an active field of research. Most explanations fall into one of two categories (or both): BP in overweight/obese people is increased either through stimulated activation of the sympathetic nervous system, or via increased sodium retention by the kidneys [14] in conclusion, this study indicated a higher prevalence of pre-hypertension and hypertension in males compared to females, although females had a higher prevalence of overweight and obesity compared to males. An association between BMI and BP was found. These results highlight that it is important to screen adolescents for both blood pressure and body weight, especially in adolescents who were found to have pre-hypertension, as they are at an increased risk of developing hypertension. This will assist in preventing any CVDs associated with hypertension. BP should be at least be screened on three separate visits to ensure the validity and address the common phenomenon of high BP prevalence.

## Figures and Tables

**Figure 1 children-07-00018-f001:**
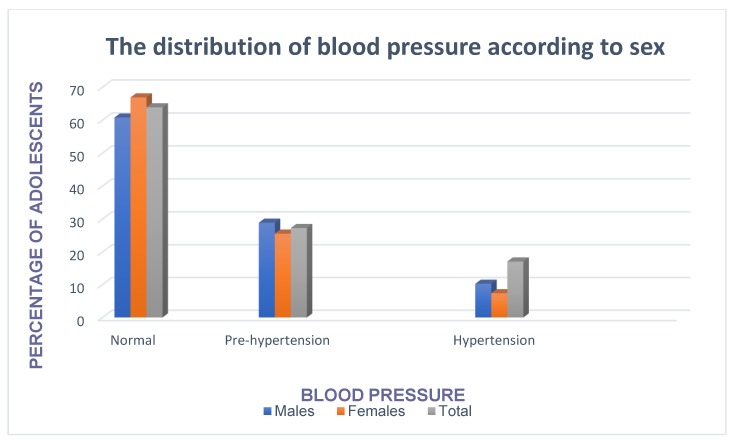
The prevalence of pre-hypertension and hypertension according to sex.

**Figure 2 children-07-00018-f002:**
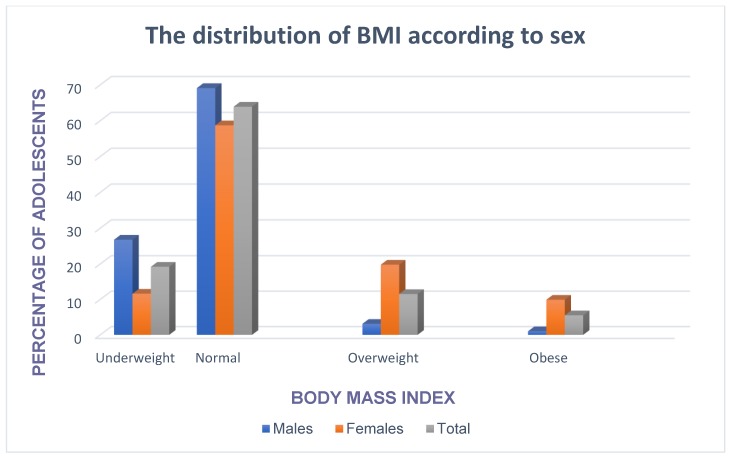
The prevalence of overweight and obesity according to sex.

**Figure 3 children-07-00018-f003:**
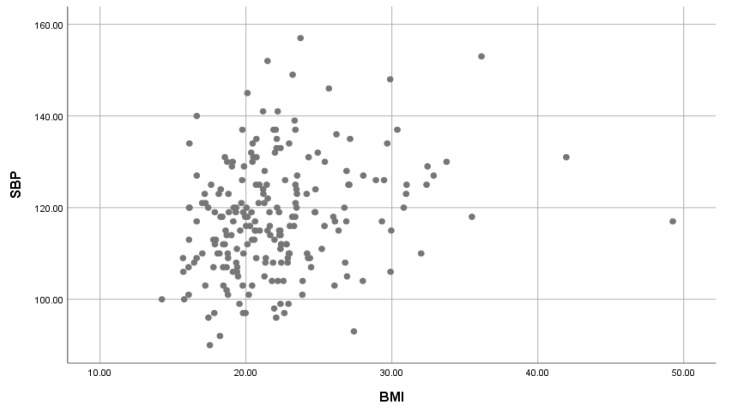
The association of BMI on SBP. BMI: Body Mass Index; SBP: Systolic blood pressure.

**Figure 4 children-07-00018-f004:**
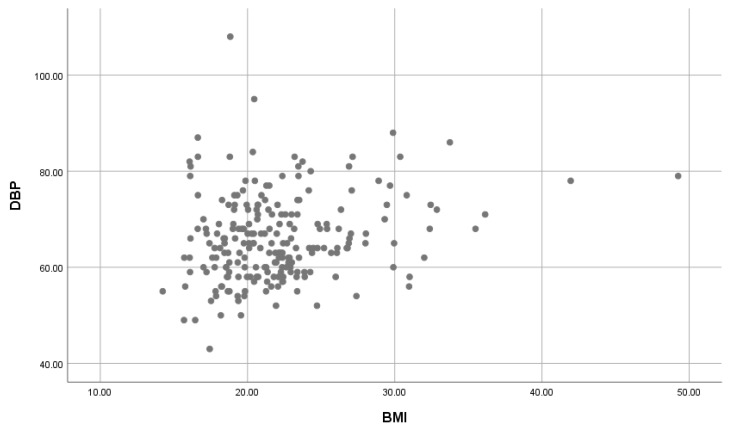
The association of BMI on DBP. BMI: Body Mass Index; DBP: Diastolic blood pressure.

**Figure 5 children-07-00018-f005:**
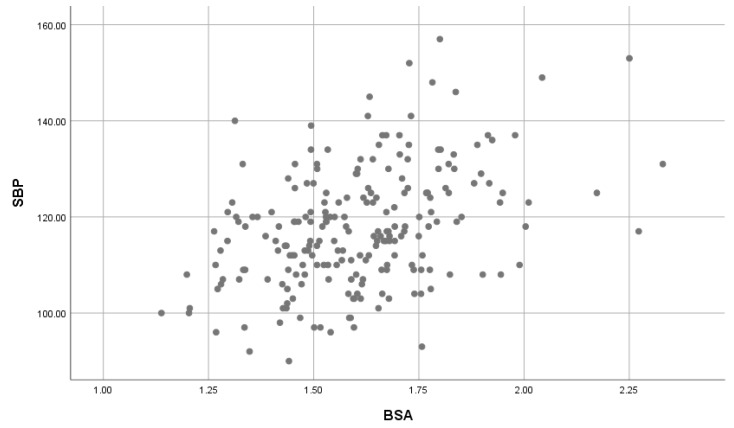
The association of BSA with SBP. BSA: Body Surface area; SBP: Systolic blood pressure.

**Figure 6 children-07-00018-f006:**
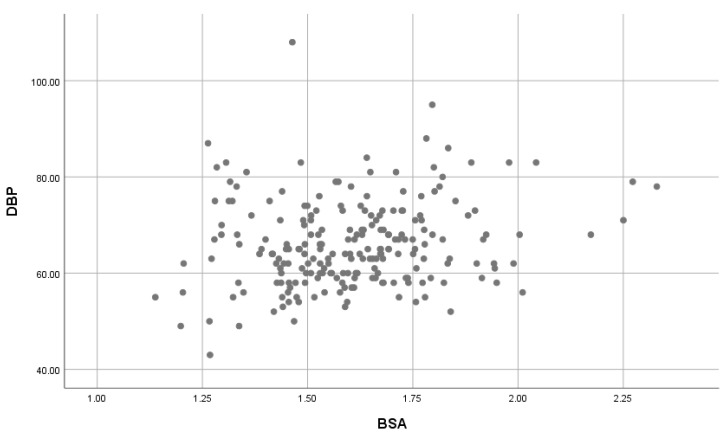
The association of BSA with DBP. BSA: Body Surface area; SBP: Systolic blood pressure.

**Table 1 children-07-00018-t001:** Heart, Lung and Blood Institute guidelines for the diagnosis of HT in children and adolescents (thirteen to seventeen).

Classification	Systolic and/or Diastolic Blood Pressure (mmHg)
Normal BP	<90th percentile
Prehypertension	90th to <95th percentile or≥ 120 mmHg
Hypertension	>95th percentile

Source adapted [10].

**Table 2 children-07-00018-t002:** Descriptive statistics of the whole study population in secondary learners aged 13–19 years in the Polokwane.

	Males (*n* = 97)	Females (*n* = 121)
	Mean ± SD	Mean ± SD
Age (years)	17.3 ± 1.7 *	16.8 ± 1.7
SBP (mmHg)	120.3 ± 13.7 *	115.7 ± 10.7
DBP (mmHg)	64.8 ± 8.8	67.0 ± 9.0 *
Weight (kg)	56.1 ± 11.6	59.4 ± 14.4 *
Height (m)	1.7 ± 0.9	1.6 ± 0.7
BMI (kg/m^2^)	20.4 ± 3.0	23.6 ± 5.2 *
BSA (m^2^)	1.6 ± 0.2 *	1.6 ± 0.2 *

* *p* < 0.05 significant difference between males and females; DBP—diastolic blood pressure; SBP—systolic blood pressure; BMI—body mass index and BSA—body surface area; *n*- is the number of participants.

**Table 3 children-07-00018-t003:** Distribution blood pressure according to body mass index.

BP	Normal	Pre-Hypertension	Hypertension	Total
BMI	*n* (%)	*n* (%)	*n* (%)	*n* (%)
Underweight	35 (16.1)	4 (1.8)	1 (0.4)	40 (18.3)
Normal	87 (39.9)	39 (17.9)	12 (5.5)	138 (63.3)
Overweight	15 (6.9)	7 (3.2)	5 (2.3)	27 (12.4)
Obese	3 (1.4)	9 (4.1)	1 (0.4)	13 (6.0)
**Total**	140 (64.3)	59 (27)	19 (8.6)	218 (100.0)

BMI: body mass index; BP: blood pressure; *n*: number of participants.

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
