# Peer review of "A Non-Invasive Investigation into the Prevalence of Higher than Normal Blood Pressure, Hypertension and the Association between Blood Pressure and Body Weight in Male and Female Adolescents in the Polokwane Local Municipality, Limpopo-South Africa: A Cross-Sectional Study"

_children, 2020, doi:10.3390/children7030018_

Round 1

Reviewer 1 Report

The manuscript entitled ‘A non-invasive investigation into the prevalence of higher than normal blood pressure, hypertension and the association between blood pressure and body weight in male and female adolescents in the Polokwane Local Municipality, Limpopo-South Africa: a cross-sectional study’ by Raphadu et al., is a study of prevalence of hypertension in 218 adolescents. To this end, the authors collected data of height, weight and blood pressure from all participants with proper exclusion criteria. The authors then performed statistical analysis for differences between males and females. The authors also tested association between BMI and blood pressure by binary logistic regression model.

Comments

Line 94, authors mention the Ref.11 as the source from which the classification was adapted. Can the authors recheck that Ref.11? It seems unrelated. The authors have presented results in two sub-categories: Characteristics of population and Effect of BMI on blood pressure. I suggest to add another category after the ‘Characteristics of population’ related to the data obtained about the differences in blood pressure and BMI. Table 5, replace B with beta. The Discussion part can be made more meaningful by further comparison (of the association of blood pressure and BMI) with data obtained on similar kind of studies in other countries. It is not entirely clear to the reader the significance of this study, as similar studies were previously reported e.g. Nkeh-Chungag et al, 2015. The manuscript has several grammatical and punctuation errors. It would be significantly better if checked by a native English speaker.

Author Response

ITEMIZED LIST OF ALL CHANGES MADE, REBUTTAL IN RESPONSE TO THE REVIEWERS' SUGGESTIONS

Tittle: A non-invasive investigation into the prevalence of higher than normal blood pressure, hypertension and the association between blood pressure and body weight in male and female adolescents in the Polokwane Local Municipality, Limpopo-South Africa: a cross-sectional study.

COMMENT

LOCATION

AMENDMENT

Reviewer 1

  1. Line 94, authors’ mention the Ref. 11 as the source from which the classification was adapted. Can the authors recheck that Ref. 11? It seems unrelated.

Materials and Methods (2.3)

Changes effected as suggested

-Inserted the related reference (10). 

2.    I suggest to add another category after the ‘Characteristics of population’ related to data obtained about the differences in blood pressure and BMI

Results

Changes effected as suggested

-A graph on the differences in the blood pressure and BMI is addressed in (Figure 1 and 2), line 147 and 149. Table 4, Line 159 also indicates the difference in BP and BMI but in percentage.

3.    Table 5, replace B with beta.

Table 5

Changes effected as suggested

-B was replace with beta as suggested, however the table has been removed.

  1. The Discussion part can be made more meaningful by further comparison (of the association of blood pressure and BMI) with data obtained on similar kind of studies in other countries.

The statements were revised and restructured as suggested. Discussion

Changes effected as suggested

5.    It is not entirely clear to the significance of this study, as similar studies were previously reported e.g. Nkeh-Chungag et al, 2015.

Aims

Changes effected as suggested

-The study was investigating the association between blood pressure and body weight, but looking into BMI and BSA; and not BMI only. As most studies did not look into BSA. I believe it’s important to investigate the association of BP looking into all different ways on measuring body weight if possible, particularly in the Polokwane region 

6.    The manuscript has several grammatical and punctuation errors. It would be significantly better if checked by a native English speaker.

Inconsistency in putting units and typographical errors were corrected as suggested

Changes effected as suggested

Reviewer 2 Report

General comments:

Raphadu and colleagues have performed the cross-sectional study among the adolescents with regard to the hypertension and body mass in the Polokwane Local Municipality of the Limpopo province of South Africa.

Comments:

1) With regard to the inclusion and exclusion criteria: was there any monitoring of smoking habits among those adolescents? Even if the smoking would not be the exclusion criterion, it would be useful to have the data about the prevalence of smoking among the study participants.

2) BMI was calculated. Why not the body surface area (BSA)? Please provide those missing data using for example the Du Bois formula. It is also important to analyze whether there is or not the association between BSA and BP.

3) Finally, the significant association between BMI and BPs was described. However, it would be interesting to present the data as a correlation graphs. Please provide the correlation graphs (BMI versus SBP, BMI versus DBP) and relevant statistical analyses (Pearson or Spearman).

Author Response

ITEMIZED LIST OF ALL CHANGES MADE, REBUTTAL IN RESPONSE TO THE REVIEWERS' SUGGESTIONS

Tittle: A non-invasive investigation into the prevalence of higher than normal blood pressure, hypertension and the association between blood pressure and body weight in male and female adolescents in the Polokwane Local Municipality, Limpopo-South Africa: a cross-sectional study.

Reviewer 2

1.    With regards to inclusion and exclusion criteria: was there any monitoring of smoking habits among those adolescents? Even if the smoking would not be the exclusion criterion, it would be useful to have data about the prevalence of smoking among the study participants.

Changes effected as suggested

-With inclusion and exclusion criteria, smoking parameter was not included in the study.

2.    BMI was calculated. Why not the body surface area (BSA)? Please provide those missing data using for example the Du Bois formula. It is also important to analyze whether there is or not the association between BSA and BP

The comment has been addressed

Changes effected as suggested

-BSA was calculated as suggested using the Du Bois formula. BSA was analyzed using Pearson correlation to check for the association between BSA and BP (SBP and DBP).

3.    Finally, the significant association between BMI and BPs was described. However, it would be interesting to present the data as a correlation graphs. Please provide the correlation graphs (BMI versus SBP, BMI versus DBP) and relevant statistical analyses (Pearson or Spearman).

The comment has been addressed.

Changes effected as suggested.

-Correlation graph were added (Figure 4 and 5), line 162 and 167.  
